# The Role of Nutrition on Thyroid Function

**DOI:** 10.3390/nu16152496

**Published:** 2024-07-31

**Authors:** Anna-Mariia Shulhai, Roberta Rotondo, Maddalena Petraroli, Viviana Patianna, Barbara Predieri, Lorenzo Iughetti, Susanna Esposito, Maria Elisabeth Street

**Affiliations:** 1Department of Medicine and Surgery, University of Parma, 43121 Parma, Italy; annashulhai@gmail.com (A.-M.S.); robertarot.97@gmail.com (R.R.); susannamariaroberta.esposito@unipr.it (S.E.); 2Paediatric Clinic, University Hospital of Parma, 43121 Parma, Italy; m.petraroli@gmail.com (M.P.); viviana.patianna@gmail.com (V.P.); 3Pediatric Unit, Department of Medical and Surgical Sciences for Mothers, Children and Adults, University of Modena and Reggio Emilia, 41124 Modena, Italy; barbara.predieri@unimore.it (B.P.); lorenzo.iughetti@unimore.it (L.I.)

**Keywords:** thyroid, thyroid hormones, gut-thyroid axis, micronutrients, microbiome, trace elements, vitamins

## Abstract

Thyroid function is closely linked to nutrition through the diet–gut–thyroid axis. This narrative review highlights the influence of nutritional components and micronutrients on thyroid development and function, as well as on the gut microbiota. Micronutrients such as iodine, selenium, iron, zinc, copper, magnesium, vitamin A, and vitamin B12 influence thyroid hormone synthesis and regulation throughout life. Dietary changes can alter the gut microbiota, leading not just to dysbiosis and micronutrient deficiency but also to changes in thyroid function through immunological regulation, nutrient absorption, and epigenetic changes. Nutritional imbalance can lead to thyroid dysfunction and/or disorders, such as hypothyroidism and hyperthyroidism, and possibly contribute to autoimmune thyroid diseases and thyroid cancer, yet controversial issues. Understanding these relationships is important to rationalize a balanced diet rich in essential micronutrients for maintaining thyroid health and preventing thyroid-related diseases. The synthetic comprehensive overview of current knowledge shows the importance of micronutrients and gut microbiota for thyroid function and uncovers potential gaps that require further investigation.

## 1. Introduction

The thyroid gland is an essential endocrine organ that regulates physiological processes that are important for maintaining health and well-being. It is responsible for the synthesis and secretion of thyroid hormones, thyroxine (T4) and triiodothyronine (T3), which are involved in the function of all systems throughout the body [1]. The function of the thyroid gland is influenced by various factors, including diet and micronutrients [2,3]. Given its important role in maintaining homeostasis, any disruption in thyroid function can have far-reaching consequences, manifesting as a spectrum of disorders ranging from hypothyroidism to hyperthyroidism and thyroid cancer. In recent years, there has been a growing interest in the effects that nutrition can have on thyroid function. Research studies have highlighted, in particular, the influence of different dietary factors and micronutrients on thyroid hormone synthesis, metabolism, and function [2,3,4,5]. Understanding these relationships is important because diet and nutrition can be changeable factors that can potentially be improved and, as a result, improve thyroid function and reduce the incidence of thyroid disorders.

Nutrition is crucial for our well-being and the proper function of all organs and systems. Macronutrients and micronutrients are vital parts of nutrition. Lipids, carbohydrates, and proteins are macronutrients that produce energy and hormones, are required to synthesize molecules, and regulate metabolic processes [6]. Micronutrients are required in trace levels for biochemical processes such as gene transcription, enzymatic and hormonal reactions, cell protection, signal transduction, and primary and secondary metabolism [6]. Micronutrients are those essential vitamins and minerals required in small quantities that have been found to be key players in supporting optimal thyroid function [4,6], from iodine and selenium to iron and vitamin D.

To date, a thyroid–gut axis [2] is well recognized. The role of microbiota is also known to be influenced by micronutrient availability, drugs, and environmental contaminants, among other factors.

This paper synthesizes current research findings highlighting the mechanisms behind the interactions among nutrition, micronutrients, microbiota, and thyroid health.

## 2. Methods

The literature search was carried out using research strings in Pubmed, Scopus, and Mendeley via MeSH, with targeted keywords. Official sites of the WHO and European international organizations were also reviewed. The identified records were imported into Mendeley, and duplicates were removed. Reviewers independently screened the remaining papers by title and abstract, considering studies published from 1986 to March 2024. The following keywords were used for the search: thyroid, thyroid hormones, diet, micronutrients, nutrition, microbiome, gut microbiota, iodine, iron, selenium, vitamin D, vitamin A, vitamin B12, zinc, copper, magnesium, environmental pollution, combined in various ways to meet the research needs. A narrative synthesis was performed to organize and interpret the search findings.

## 3. Micronutrients and Thyroid Function

The overall effects of known nutrients and micronutrients on thyroid function are reported in Figure 1.

### 3.1. Iodine

Iodine plays an essential role in the synthesis of thyroid hormones (TH). It is absorbed in the small intestine, primarily in the form of iodide ions (I-), transported via the bloodstream to the thyroid gland, where it is actively taken up by thyroid follicular cells [7]. Within the thyroid follicular cells, iodine is enzymatically incorporated into tyrosine residues on thyroglobulin, a large glycoprotein synthesized by the thyroid gland. This process results in the formation of monoiodotyrosine (MIT) and diiodotyrosine (DIT), which then undergo further iodination to produce triiodothyronine and thyroxine, the active thyroid hormones [7]. These iodinated tyrosine residues are part of the thyroglobulin molecule, and when the hormone is required, thyroglobulin is broken down, releasing T3 and T4 into the bloodstream (Figure 1). Therefore, adequate iodine intake is necessary for maintaining normal thyroid function. The European Food Society Authority (EFSA) proposes age-specific intakes of iodine ranging from 70 µg/24 h for children to 150 µg/24 h at the end of adolescence [8,9] (Table 1). These are intakes calculated to ensure a urinary iodine concentration (UIC) ≥ 100 µg/L, which is the cut-off associated with the lowest prevalence of iodine-deficient goiter in school-aged children [10] (Table 1). It is calculated that in healthy children, the body contains around 15 to 20 mg of iodine, with the majority found in the thyroid gland.

Iodine deficiency in pregnancy can severely impact the cognitive development of offspring, as both maternal and fetal thyroid functions are affected. It is well known that in areas with severe and chronic iodine insufficiency, hypothyroidism can occur in both mothers and fetuses from the early stages of gestation, leading to irreversible brain damage, mental retardation, and neurological abnormalities [11,12]. The most severe form of in-utero deficiency is congenital hypothyroidism (formerly categorized under various “cretinism” labels) [13]. A meta-analysis evaluated iodine supplementation during pregnancy in areas with mild-to-moderate iodine deficiency; a lack of robust randomized controlled trial (RCT) evidence, with many observational studies being of poor quality, was described. It was concluded that maternal prepregnancy iodine status might influence thyroid adaptation during pregnancy, with mixed effects on thyroid volume and function. However, current evidence is insufficient to strongly support iodine supplementation recommendations in mildly-to-moderately iodine-deficient pregnant women (median UIC < 100 µg/L). Further research is needed to clarify the benefits and potential risks of iodine supplementation during pregnancy with consideration of maternal intrathyroidal iodine status [14].

Recent studies have investigated the impact of mild-to-moderate iodine deficiency during pregnancy on children’s outcomes, and in addition, in countries such as Tasmania, maternal iodine deficiency was associated with lower literacy and numeracy scores in children at 9 years of age [15,16]. Even in postnatally iodine-rich environments, language and spelling outcomes remain lower if iodine deficiency during pregnancy has occurred. In a study from the UK, maternal iodine deficiency during the first trimester of pregnancy was associated with suboptimal verbal Intelligence Quotient (IQ) and reading scores in their children at 8 and 9 years of age, respectively [16,17].

Excess iodine consumption can also have a negative impact on thyroid function. Normally, large iodine dosages cause a transitory shutdown of thyroid hormone production (the acute Wolff-Chaikoff effect). Continuous exposure to high iodine levels causes down-regulation of the sodium iodide symporter (NIS), which transports iodine into the thyroid, enabling thyroid hormone production to continue; this is known as an escape from the acute Wolff-Chaikoff Effectf [18]. Failure of this effect can lead to iodine-induced hyperthyroidism or hypothyroidism, especially in individuals with pre-existing thyroid conditions. The fetus is particularly vulnerable to hypothyroidism, as the ability to fully escape this effect does not develop until about 36 weeks of gestation. Maximum safety limits for iodine intake in pregnant women, breastfeeding women, and infants, however, have not yet been comprehensively defined [16].

Universal salt iodization is considered the most effective method to ensure sufficient iodine intake in the population [12,19]. However, it has not been universally adopted, not implementing this strategy. Even in areas where salt iodization programs are in place, pregnant women who require higher iodine levels may still have insufficient intake. This has led various medical societies and government bodies, such as the American Thyroid Association, the Endocrine Society, the U.S. Teratology Society, the American Academy of Pediatrics, and the European Thyroid Association, to recommend iodine supplementation for pregnant, lactating, or women planning pregnancy [16,20]. The World Health Organization (WHO) provides iodine supplementation guidelines based on household iodized salt consumption [19]. In regions where less than 90% of households consume iodized salt and the median urinary iodine concentration is below 100 μg/L, iodine supplementation in the form of potassium iodide is advised, with an intake target of 250 μg/day. Alternatively, in severely iodine-deficient populations where daily supplementation may be impractical, an annual 400 mg dose of iodized oil supplement can be considered. Ideally, iodine supplementation should be started at least 3 months before conception to ensure that maternal thyroid iodine stores are sufficient for the entire pregnancy [12,16,20].

The role of iodine during early childhood is crucial for brain development and TH synthesis. Adequate iodine intake is particularly critical during the weaning period, as infants transition from exclusive breastfeeding to a mixed diet [21]. While breastfeeding is an excellent source of iodine for infants, the amount in breast milk largely depends on the mother’s iodine status. In the United States, the iodine content of breast milk has been shown to vary geographically, with some regions experiencing mild-to-moderate deficiency. This can be attributed to factors such as decreased iodine intake in maternal diets, decreased iodine content in cow’s milk if taken by the mothers during breastfeeding, or the absence of iodine supplementation during pregnancy [21]. A 2018 study conducted in a non-iodine-deficient region showed that although there is a correlation between Maternal Body Iodine Content (BMIC) and the iodine status of infants, the feeding type (exclusive breastfeeding vs. partial formula feeding) does not influence the UIC values of infants and mothers in analyses adjusted for BMIC [22]. The introduction of complementary nutrients during weaning is an opportunity to enhance iodine intake. However, studies have suggested that the iodine content of many infant foods is inadequate to meet the recommended daily intake. Commercially available baby food, including fruit, vegetables, and cereals, often lacks sufficient iodine levels. Moreover, organic baby foods, which may appeal to health-conscious parents, tend to have even lower iodine content. Low iodine intake during weaning can have long-term consequences in cognitive development and thyroid function in infants. Studies have shown that infants with insufficient iodine intake may exhibit lower IQ scores and an increased risk of thyroid-related disorders [13,22]. Strategies should be implemented at both the individual and public health levels to address the iodine nutrition gap in weaning infants. Healthcare providers can play a crucial role in educating parents about the importance of iodine-rich foods and the potential need for iodine supplementation. Manufacturers can also contribute by fortifying infant foods with iodine to ensure an adequate intake for weaning infants.

During adolescence, deficient iodine intake can lead to nodular goiter, while excess intake or iodine fortification can increase the risk of thyroid autoimmunity. The mechanisms underlying how iodine leads to thyroiditis remain unclear, and various immunological and non-immunological theories have been proposed. Among these, it is hypothesized that excess and accumulation of iodine in thyroid cells can generate high oxidative stress and cause cellular damage. This can lead to the production and secretion of cytokines and chemokines, recruiting lymphocytes to the thyroid, where they encounter thyroid auto-antigens, such as thyroglobulin. Excess iodine can alter the conformation of thyroglobulin, making it more easily recognized by immune system cells [23]. This can ultimately lead to the development of pathological hypersensitivity to thyroid auto-antigens, with subsequent development of thyroiditis [23,24]. In China, after 3 years of implementing salt iodization, the prevalence of autoimmune thyroid diseases varied based on iodine intake levels: 0.5% in mildly deficient areas, 1.7% in areas with more-than-adequate iodine, and 2.8% in regions with excessive-high consumption [25]. Meanwhile, in Denmark, which previously experienced mild-to-moderate iodine deficiency, iodine status significantly improved after 5 years of mandatory iodine fortification of salt, transitioning from deficient to optimal iodine intake levels also leading to a decreased prevalence of autoimmune thyroid diseases [26]. Therefore, special attention must be paid to nutritional intake once adequate iodine uptake (aiming for ioduria >100 µg/L, a value associated with minimal goiter incidence in the growing population) has been ensured.

Nowadays, with global gastronomic influences significantly altering adolescent eating habits, this becomes even more crucial. In a study involving Italian children and adolescents [27], iodine requirements were met primarily through excessive salt intake (>10.2 g/day). This not only highlighted the issue of excessive salt consumption but also emphasized that iodized salt remains underutilized in industrial processes and catering services [27]. Furthermore, this study [27] provided estimated values for iodine intake from unsupplemented food in different age groups: 44 µg/1000 Kcal for children, 45 µg/1000 Kcal for teenage girls, and 35 µg/1000 Kcal for teenage boys. It is commonly assumed that certain foods, like those from the Brassicaceae family, cassava, and millet, contribute to iodine malabsorption [27]. However, this is only significant when consumed in large quantities under conditions of iodine deficiency. Of particular interest is the mechanism involving thiocyanates, which competitively inhibit iodine uptake by thyrocytes and facilitate the efflux of intrathyroidal iodine [28].

Notably, cigarette smoke contains substantial amounts of thiocyanates, explaining the association between smoking and goiter [28]. This association is even more pronounced in cases of obesity [28]. This underscores the importance of reinforcing prevention campaigns targeting smoking and obesity within this age group [10].

### 3.2. Selenium

Selenium (Se) is an essential trace element crucial for the thyroid system. The human body contains approximately 14 mg of Se [29], and it is an integral component of the enzymes glutathione peroxidase (GPx) [30] and iodothyronine deiodinases, where it is incorporated as selenocysteine. Selenium-dependent GPx catalyzes the breakdown of hydrogen peroxide (H_2_O_2_), shielding most tissues from oxidative stress and cell damage [31]. In the thyroid gland, H_2_O_2_ is produced in response to thyrotropin (TSH) stimulation and is required for thyroglobulin iodination by the enzyme thyroid peroxidase (TPO), which leads to T4 production. Excess H_2_O_2_ has been demonstrated to inactivate thyroid peroxidase in vitro, suggesting that its accumulation could also affect TH [32] (Figure 1).

Regarding the prenatal period, a significant RCT involving pregnant women positive for anti-thyroid peroxidase antibody (TPOAb), demonstrated that selenium supplementation led to a notable reduction in TPOAb levels during and after pregnancy [20,33]. This supplementation also reduced the incidence of postpartum thyroid disease and permanent hypothyroidism [33]. At variance with the study, another RCT examining the impact of selenium supplementation on autoimmune thyroid disease in pregnancy did not find any reduction in TPOAbs between the Se and placebo groups. However, this study had lower baseline TPOAb concentrations, mothers received a lower selenium dose (60 µg/d), and the study was overall underpowered [34]. There is a clear need for high-quality, well-powered RCT focusing on TPOAb-positive pregnant women to validate the findings of the Italian study.

In areas with combined selenium and iodine deficiencies, selenium deficiency may reduce fetal hypothyroidism by reducing maternal T4 metabolism, offering protection to fetal brain development. However, it is hypothesized that after birth, the iodine-deficient thyroid gland, which is hyperplastic and hyperstimulated, generates an excess of free oxygen radicals that, in addition to reduced GPx activity, could trigger the destruction of the thyroid gland. This intriguing hypothesis awaits formal validation and is not fully corroborated by epidemiological evidence [35].

Adequate iodine levels should be ensured before administering selenium compounds to prevent unintended loss of iodinated TH [13].

Selenium deficiency has been associated with various thyroid disorders, including hypothyroidism, subclinical hypothyroidism, thyroid cancer, and autoimmune thyroid diseases, including HT and GD [36]. In childhood and adolescence, there are few studies. For instance, some research has focused on children and adolescents who have phenylketonuria (PKU). Van Bakel et al. have found significantly higher fT4 and fT3 concentrations in children with PKU compared to control subjects [37]. Furthermore, Calomme et al. studied the effect of selenium supplementation on TH concentrations in ten severely deficient subjects: Se supplementation led to a significant increase in plasma selenium concentrations and GPx activity, and this was associated with a notable decrease in plasma T4 (−16%) and rT3 (−29%) compared to initial values. Discontinuing Se treatment resulted in a significant increase in T4 and rT3 levels, which returned to their initial values. Plasma TSH and T3 concentrations were unaffected by selenium status [38]. Both studies demonstrated a negative connection between plasma selenium and plasma T4 and rT3 levels [37,38]. However, these investigations found no evidence that selenium impacted TH feedback at the pituitary level or produced clinically detectable hypothyroidism. These findings are consistent with those from animal research [39,40]. Prolonged and isolated selenium deprivation in rodents results in a complete loss of liver 5′-deiodinase (5′DI) activity and an increase in circulating T4, with little to no change in T3 or TSH concentrations. Interestingly, selenium insufficiency is linked to higher blood rT3 concentrations in young rats but not in adult males or non-pregnant females. This might represent a gradual increase in 5′DI expression early in life. The anticipated significant increases in cerebrocortical and pituitary 5′DII activities in response to hypothyroidism, as well as thyroid GPx an’ 5′DI and pituitar’ 5′DI activities, are mostly unaffected by selenium deficiency in rats. This supports the absence of biochemical or clinical signs of hypothyroidism in selenium-deficient PKU patients [35].

In humans, the impact of selenium supplementation (without iodine supplementation) on thyroid function was evaluated in 23 school-age children residing in an area severely deficient in both iodine and selenium. Selenium supplementation elevated plasma selenium levels and erythrocyte GPx activity but led to a substantial 30–50% reduction in serum T4 and free T4 concentrations, transitioning from normal to hypothyroid levels. Plasma TSH and T3 concentrations stayed within the euthyroid range at baseline and remained unaffected by selenium supplementation. This suggests that in iodine-deficient regions, addressing concurrent selenium deficiency without iodine supplementation heightens the conversion of T4 to T3 in the periphery [41].

A recent observational cohort study on the adult population documented an increased incidence of Hashimoto’s thyroiditis (HTs) after a 6-year follow-up in 1254 individuals with low selenium status. This study provided epidemiological support for a strategy to prevent autoimmune thyroiditis by ensuring adequate Se nutritional intake, especially in women [42].

Various systematic reviews/meta-analyses have examined the effects of selenium treatment in autoimmune thyroid diseases and/or Hashimoto’s disease. One meta-analysis of 16 trials concluded that selenium supplementation reduced TPOAb levels, especially in patients already on L-T4 treatment [43]. The systematic review from 2017 identified only three studies showing some improvement in the quality of life and thyroid echogenicity on ultrasound in patients treated with levothyroxine replacement therapy with selenium supplementation. However, it was not possible to synthesize them in a meta-analysis due to the small sample sizes [44]. A recent meta-analysis revealed a substantial decrease in TSH levels and anti-TPO antibodies in euthyroid and subclinical hypothyroid individuals with HTs without hormone replacement therapy (THRT) after selenium supplementation [45,46]. These findings suggest a potential benefit of selenium supplementation for autoimmune thyroid conditions. However, larger, more comprehensive studies are required to conclusively confirm these effects. To reduce the risk of selenium poisoning, physicians should follow recommended selenium doses (80–400 μg/day) and avoid extended high-dose supplementation. Potential adverse effects should be considered, especially in non-selenium-deficient people [45].

### 3.3. Iron

Iron (Fe) is the third essential trace element important for the normal biosynthesis and function of THs. An adult human body contains about 4 g of iron (0.005% of body weight). Most of this iron is found in hemoglobin and myoglobin, but it is also present in various cytochromes and other heme-containing proteins. Fe is the central atom in the active sites of these proteins, including TPO, which is uniquely expressed by thyroid cells and is related to myeloperoxidase and lactoperoxidase [13] (Figure 1). TPO is a crucial multifunctional enzyme in the synthesis of thyroid hormones [47].

During pregnancy, there is an increased demand for iron to support the expansion of maternal blood volume and meet the needs of the developing fetus. Previous studies have shown that iron deficiency can be a risk factor for thyroid disorders during pregnancy [48]. The effect of this insufficiency on TH may be correlated with changes in the activity of TPO or the reduction in the conversion of T4 to T3 by influencing the activity of thyroxine deiodinase [49]. Several studies have observed lower levels of fT3 and fT4 in pregnant women with iron deficiency [50,51,52]. Iron and hemoglobin have been found to be positively correlated with fT3 and fT4 but negatively correlated with TSH, suggesting that the fT4 level might change with iron status. Regular monitoring of iron status and thyroid function, along with appropriate supplementation and interventions, is essential to mitigate potential risks associated with deficiencies or dysregulation during pregnancy [53].

Iron deficiency and anemia are associated with hypothyroidism, likely due to the reduced biosynthesis of the hemoprotein TPO [54]. However, the interplay between iron and TH is bidirectional, as TH directly stimulates erythropoiesis, predominantly through the TR-alpha receptor [55,56]. Iron supplements and improving iron status in areas with endemic deficiency increased TH biosynthesis, as demonstrated in children [57] and anemic patients (especially women) [5].

The association between iron deficiency and thyroid autoimmunity has not been firmly established. Many patients with HTs have iron deficiency due to comorbidities such as autoimmune gastritis, which leads to reduced iron absorption, or celiac disease, which results in iron loss [58]. In a recent retrospective study, TSH, TPOAb, and anti-thyroglobulin antibody (TgAb) levels were significantly higher, whereas hematocrit, mean corpuscular volume, hemoglobin, ferritin, and iron were significantly lower in 180 female patients with positive thyroid autoantibodies compared to 81 healthy controls [59]. Combined treatment with T4 and sufficiently bioavailable iron supplements may be necessary, and clinical studies have shown better effects than those obtained with each factor taken individually [13,60], further investigations of iron status on the efficacy and absorption of thyroid hormone replacement therapy.

### 3.4. Vitamin D

Vitamin D receptors are present in the thyroid gland, indicating a potential role in thyroid function [61]. Vitamin D modulates the immune system by boosting the innate immune response and inhibiting the adaptive immune system [62] (Figure 1). Vitamin D’s capacity to suppress the adaptive immune system improves immune tolerance and appears to help with a variety of autoimmune disorders [62]. Pleiotropic activities have been identified through preclinical and observational research, indicating a positive role in thyroid disease treatment. However, few equivocal causal correlations and a few intervention studies have been documented thus far, particularly in the juvenile population. Therefore, the preventative and therapeutic potential of vitamin D or its analogs in thyroid illnesses is still being debated.

Animal studies have provided evidence for the role of vitamin D in autoimmune thyroid diseases.

Active vitamin D (25-hydroxyvitamin D—25(OH)D) crosses the placenta, and the mother’s adequate vitamin D status is important for the health of both the mother and the child [63]. There are significant regional and ethnic differences in the prevalence of vitamin D insufficiency in pregnancy, ranging from 18 to 84% [63,64,65]. There is limited data regarding the relationship between vitamin D status and thyroid autoimmunity during pregnancy. In some studies, the prevalence of TPOAb was higher in the first trimester [66,67].

In childhood/adolescence, clinical studies have reported low vitamin D levels in autoimmune thyroid disease (AITD), indicating an association between vitamin D deficiency and thyroid autoimmunity [68]. Otherwise, the association between vitamin D and HTs remains controversial. Some studies, including observations from the Croatian Biobank of HTs patients and other comparative studies, have not detected an association between vitamin D levels and the prevalence of HTs [69,70,71]. Other studies have observed higher rates of vitamin D deficiency in children with HTs (inverse correlation between 25(OH)D and TPO- Ab levels) [68]. This difference was more evident in patients with overt hypothyroidism compared to those with subclinical hypothyroidism. Furthermore, significant negative correlations were found between serum levels of 25(OH)D and patient age, disease duration, BMI, and TPOAb, TgAb, and TSH levels. The authors proposed low serum vitamin D to be significantly associated with autoimmune thyroid disease, but this was not an independent risk factor for the progression of the condition [72]. It is worth noting that recently, studies evaluating whether vitamin D supplementation is beneficial for autoimmune thyroid disease have been published. After 4 months of oral vitamin D3 supplementation (1200–4000 IU/day) in 186 deficient patients, there was a significant decrease (20.3%) in serum TPOAb levels [73,74]. In a study by Taheriniya et al., which examined 42 studies involving patients with AITD, it was found that vitamin D deficiency correlates with the onset of AITD, including HTs and hypothyroidism [75]. A relationship between vitamin D and Graves’ disease has only been reported in older people [75]. The Vitamin D Receptor (VDR) and 1α-hydroxylase are present in various immune cells including T and B lymphocytes, dendritic cells, neutrophils, and monocytes [73,74]. This allows these cells to produce calcitriol, the active form of vitamin D3. It inhibits the production of proinflammatory cytokines such as IL-6, IL-8, IL-9, IL-12, IFN-γ, and TNF-α, while enhancing the production of anti-inflammatory cytokines such as IL-10, IL-5, and IL-4. Overall, vitamin D is considered to have an anti-inflammatory effect [76,77].

Low vitamin D status may be a consequence rather than a cause of thyroid diseases, and it can be influenced by various factors such as low intake of vitamin D, malabsorption, lack of sun exposure, or reduced outdoor activity [78]. The last meta-analyses showed an increased risk of thyroid cancer in subjects with vitamin D deficiency [79]. Study design limitations, inter-assay and inter-laboratory variability in vitamin D measurements, and heterogeneity of the study populations also contribute to the conflicting results [80]. The preventive and therapeutic potential of vitamin D or its analogs in thyroid diseases remains a topic of debate, and ongoing research will provide insight into their efficacy and safety as therapeutic tools.

A recent meta-analysis showed that vitamin D supplementation significantly decreases TPOAb and TgAb titers in patients undergoing thyroid therapy, leading to improved thyroid function with reduced TSH levels and increased FT3 and FT4 levels [81]. Vitamin D supplementation at a dosage of 1500 to 2000 IU/d for more than 12 weeks resulted in more significant decreases in TPOAb levels and higher increases in FT3 and FT4 compared to shorter treatments. Vitamin D administration may reduce TPOAb and TgAb titers by improving thyroid function through immunomodulation. Therefore, vitamin D supplementation might be beneficial for patients with HTs [81]. However, further research is needed to understand the specific mechanisms behind these effects and long-term benefits.

### 3.5. Zinc

Zinc (Zn) is a trace mineral that contributes to gene expression and cell growth and serves as a cofactor for numerous enzymes involved in various physiological processes, including thyroid hormone synthesis and metabolism [82,83]. The most common nutrition rich in zinc are oysters and fish, legumes, nuts, red meat, whole grains, and dairy [6]. Within the thyroid gland, zinc plays a critical role in the activity of TPO, an enzyme essential for the synthesis of thyroid hormones. TPO catalyzes the iodination of thyroglobulin, a protein precursor of thyroid hormones, as well as the pairing of iodotyrosine residues to form T4 and T3 [84]. Experimental studies in vitro and in vivo have provided evidence that zinc behaves as an antioxidant, reducing the oxidation of macromolecules like DNA/RNA and proteins. It also mitigates the inflammatory response, leading to a decrease in the formation of reactive oxygen species (ROS) [85]. Autoimmune disorders are linked to abnormal zinc levels and can lead to disrupted signal transduction, affecting the immune response, cell function, and differentiation. Zinc deficiency can weaken both the innate and adaptive immune responses. Additionally, T cell activation and differentiation into specific subpopulations (Th1, Th2, Th17, and Treg) significantly influence zinc balance [86,87,88].

Zinc deficiency can impair thyroid hormone receptor function, affecting their ability to bind thyroid hormones effectively and regulate gene expression, leading to disruptions in thyroid hormone signaling pathways [89], may influence decreased thyroid hormone synthesis, and potentially contribute to hypothyroidism development [90,91]. Many studies and systematic reviews report on the association between Zn deficiency and hypothyroidism [90,92,93,94]. Acquired zinc deficiency can cause another common symptom in patients with hypothyroidism, which is hair loss [90]. The loss of pigment, dryness, brittleness, and hair loss appear due to Zn deficiency as a cofactor of metalloenzyme. Some researchers suggest using a diet that is rich in zinc to achieve euthyroidism, which can be beneficial for alopecia [90,95,96,97]. However, further randomized clinical trials are required for understanding the role of zinc supplementation on hair loss in hypothyroid patients. Nevertheless, there is limited data about the zinc–hyperthyroidism relation, which is controversial and requires further investigation. Some studies have demonstrated zinc’s influence on papillary thyroid cancer and follicular carcinoma [98,99]. However, a recent Mendelian randomized study found no evidence of this influence [100], indicating the need for further understanding.

Zinc deficiency has been linked with thyroid enlargement in children and adults. The study of 68 school-age children showed a positive inverse correlation between zinc concentration and thyroid size [101]. Researchers hypothesized the main cause of this is Zn loss and impaired T3 action and T3 binding to its nuclear receptor [101]. The same results were found in adults, where Zn concentration was negatively correlated with thyroid volume [91], and adults with nodular goiter, especially in iodine deficiency regions [102]. Subjects with low zinc levels have shown a positive correlation with autoimmune thyroid disease [91]. However, this finding is controversial, as an investigation of Zn level differences between women with Hashimoto thyroiditis and healthy controls did not show any differences [103], highlighting further need for investigation of the association of HTs and zinc relations.

It is worth noting that recently, the RCT of daily zinc supplementation for 12 weeks on serum thyroid auto-antibody levels in children and adolescents with autoimmune thyroiditis has found no difference in thyroid levels, TPOAb, TgAb, and oxidative stress markers [104]. Interestingly, despite the absence of thyroid hormones and auto-antibodies changes, levothyroxine dose requirements significantly increased in the control (placebo) group compared to the group with zinc supplementation [104], but further long-term trials are required. So, zinc supplementation might be useful in hypothyroid patients [104,105].

## 4. Other Micronutrients: Copper, Magnesium, Vitamin A, and Vitamin B12

### 4.1. Copper

Copper (Cu) is a trace mineral that can be found in organ meats, fish and shellfish, seeds and whole grains, chocolate, and leafy greens [6]. It is involved in several key processes related to thyroid hormone synthesis and regulation. The recommended dietary allowance (RDA) is approximately 1 mg/d for both adult men and women [6]. Copper is a cofactor for tyrosinase and is involved in the conversion of inactive T4 to the biologically active T3, necessary for the synthesis of TPO, a precursor to thyroid hormones, and the subsequent coupling of iodotyrosine residues to form T4 and T3 [6,89]. Moreover, Cu has pro-oxidant and antioxidant characteristics, and its imbalance leads to oxidative stress and potentially may lead to thyroid dysfunction [106]. As a pro-oxidant, Cu can participate in Fenton-like reactions, producing highly reactive hydroxyl radicals from hydrogen peroxide. These hydroxyl radicals are potent oxidizing agents that can damage cellular components, including lipids, proteins, and DNA. Antioxidant actions occur through Cu’s role in superoxide dismutase for oxidative stress mitigation and copper-dependent enzymes (cytochrome c oxidase and ceruloplasmin) [106]. Cu is associated with the regulation of body calcium levels, which in turn prevents the over-absorption of T4 in blood cells and is essential for supporting optimal thyroid function and hormone regulation [89]. A deficiency of Cu is related to subclinical hypothyroidism [107] or hypothyroidism [108]. However, a large USA survey showed that Cu elevation is related to higher levels of fT4 and TT4 in males and TT3 and TT4 in females [109], which requires further studies. There are limited studies about Cu and HTs. Szczepanik et al. (2021) have not found significant differences between Cu levels in patients with HTs and controls [103]. In contrast, Rasic-Milutinovic et al. (2017) defined higher Cu levels in patients with HTs [109]. In this single study, researchers hypothesized that elevated serum selenium levels and reduced copper levels in patients with HTs and overt hypothyroidism might result in lower L-thyroxine dosages or reaching a euthyroid state with decreased free T4 due to the direct impact of the copper-selenium ratio on redox balance and as a result of thyroid health [109]. Moreover, excessive copper intake can be detrimental to thyroid function and indicate proliferative process progression in the thyroid gland. Thus, the meta-analysis showed that higher Cu was found in patients with thyroid cancer in China and Turkey. As copper is a cofactor in tumor angiogenesis, and its high levels influence tumor growth [110], Cu agents potentially can be used for BRAFV600E mutation-positive malignancies treatment and tumors resistant to BRAFV600E and MEK1/2 inhibitors [111].

### 4.2. Magnesium

Magnesium (Mg) is a mineral that can be found in leafy green vegetables, legumes, nuts, seeds, milk, whole grains, etc. [6]. It is involved in various aspects of thyroid function, as it is required for the activation of adenosine triphosphate (ATP) and DNA replication and transcription. Magnesium acts as a cofactor for several enzymes and enzymatic reactions and is also involved in thyroid hormone metabolism. It can indirectly influence deiodination, which catalyzes the conversion of T4 to the more active T3 form [6,112]. Additionally, it is involved in the regulation of thyroid hormone receptor sensitivity as a second messenger, which affects the receptivity of target tissues to thyroid hormones and balances oxidative phosphorylation [113]. Magnesium shortage impacts the bioavailability and tissue distribution of selenium, resulting in reduced levels [114]. Experimental studies showed an increase in follicular-colloidal index, amount of thyrocytes in the follicle and resorption vacuoles in the colloid, and activation of thyroid gland synthetic activity by ingestion of magnesium chloride [115]. Mg deficiency is common in the general population and has been associated with impaired thyroid function, metabolic disorders, and carcinogenesis. Studies involving 1257 subjects demonstrated elevated risks of TgAb positivity and higher HTs and hypothyroidism prevalence in case of severe Mg deficiency (0.55 mmol/L), suggesting that blood magnesium levels should be investigated in individuals with autoimmune thyroiditis and hypothyroidism [116]. Hypothesized that magnesium deficiency, in combination with coenzyme Q10 and selenium and inefficient oxidative phosphorylation, leads to mitochondrial dysfunction and potentiates hyperthyroidism development [114]. Moreover, magnesium deficiency has an impact on cancerogenesis by linking with inflammation and/or free radicals, which can cause DNA oxidative damage and cancer formation. The cross-sectional study analyzing 5709 who underwent thyroidectomy found that patients with papillary thyroid cancer had lower serum magnesium levels within the common range compared to benign nodules [117]. Notably, magnesium level > 935 μmol/L was found to be a potential independent protective factor against papillary thyroid cancer [117]. Based on this, more studies should be done to find Mg’s possible potential protective role against thyroid cancer and establish guidelines to prevent cancerogenesis. Additionally, comparing premenopausal and postmenopausal women, researchers have defined a significant decrease in magnesium in postmenopausal women [112]. There is an indirect relation between Mg and thyroid hormone levels, particularly with TSH levels. Hypomagnesemia development can be one of the factors contributing to the development of thyroid disorders in the postmenopausal period [112]. However, a high dose of magnesium can increase thyroid activity [110]. Animal studies on albino Wistar strain rats showed that chronic high magnesium intake leads to thyroid disruption as it elevates thyroid peroxidase and Na(+)-K(+)-ATPase, changes iodothyronine 5′-deiodinase type I activities, increases serum T4, tT3, and TSH, resulting in lowering fT3 and T3 and enlarging of the thyroid gland [118,119]. Therefore, a balanced level of Mg, by diet or supplementation intake, may be beneficial for supporting thyroid health and overall well-being.

### 4.3. Vitamin A

The relationship between vitamin A and thyroid function is a topic of considerable scientific interest due to the intricate interplay between these two essential elements for human health. Vitamin A, a fat-soluble micronutrient, plays critical roles in various physiological processes, including vision, immune function, reproduction, and cellular growth and differentiation [120]. Within the context of thyroid function, emerging evidence suggests that vitamin A status can profoundly influence thyroid gland homeostasis through multiple mechanisms. Vitamin A deficiency (VAD) has been implicated in thyroid gland impairment, particularly when coupled with iodine deficiency [83,121,122]. Animal studies utilizing mouse models have provided valuable insights into the effects of VAD on thyroid function. These studies have demonstrated that VAD can lead to reduced uptake of iodine by the thyroid gland, impaired synthesis of thyroid hormones, and decreased production of thyroglobulin [123]. Consequently, these disruptions in thyroid hormone synthesis and secretion can contribute to thyroid hypertrophy, goiter formation, and alterations in intrathyroidal hormone levels [83,122]. Moreover, studies have highlighted the role of vitamin A in modulating the sensitivity of thyroid cells to TSH, a key regulator of thyroid function. Specifically, vitamin A has been shown to influence the expression of genes involved in TSH receptor signaling pathways, thereby affecting thyroid hormone production and secretion [83,122,124]. Additionally, vitamin A status appears to modulate the peripheral metabolism of thyroid hormones, including the conversion of T4 to the biologically active T3 hormone [119,125]. Notably, the interaction between vitamin A and thyroid function extends beyond hormonal regulation to encompass the role of transport proteins such as transthyretin (TTR). Vitamin A binds to retinol-binding protein (RBP) to form a complex that interacts with TTR, a carrier protein for both retinol and thyroid hormones in the bloodstream. Studies have suggested that changes in vitamin A status can influence the binding capacity and affinity of TTR for thyroid hormones, potentially impacting their transport and distribution in the body [126]. In regions with endemic malnutrition and micronutrient deficiencies, investigations into the effects of vitamin A supplementation on thyroid function have yielded diverse findings [127,128]. While some studies have reported improvements in thyroid volume and hormone levels following vitamin A supplementation, others have shown mixed or inconclusive results [127,128]. Factors such as the severity of VAD, the presence of concomitant ID, and individual variations in response to supplementation may contribute to the variability observed in these studies.

Moreover, vitamin A plays a crucial role in modulating immune functions, particularly through the modulation of regulatory T cells (Treg) and various cytokines [129]. Dysregulation of immune function and alterations in gut microbiota composition have been implicated in the pathogenesis of AITD. Polymorphisms in genes like CYP26B1 and RAR are associated with HTs severity and Th17 polarization, indicating a potential role for vitamin A in modulating immune responses in this condition [129]. Dysbiosis in HTs affects nutrient absorption and contributes to autoimmunity, suggesting a regulatory role for vitamin A in maintaining gut barrier integrity and potentially influencing HTs pathogenesis [129,130,131]. In summary, the relationship between vitamin A and thyroid function is complex and multifaceted, involving intricate molecular interactions and physiological pathways. While considerable progress has been made in elucidating the role of vitamin A in thyroid homeostasis, many questions remain unanswered. Future research efforts aimed to find an association between vitamin A, RBP, and TTR, and their impact on the transport and distribution of thyroid hormones in the body and the potential of vitamin A supplementation in patients with thyroid disorders.

### 4.4. Vitamin B12

Vitamin B12, also known as cobalamin, plays a crucial role in thyroid function and overall metabolic health. It is essential for the synthesis of thyroid hormones and metabolism. Dietary sources rich in vitamin B12 include animal-based foods such as meat, fish, eggs, and dairy products [132]. Vitamin B12 deficiency also contributes to AITD, such as Hashimoto’s thyroiditis, as it plays a role in immune function and regulation. A recent study demonstrates that the level of vitamin B12 shows a significant correlation with AITD [133]. Thus, assessing the concentration of vitamin B12 in patients with AITD is essential, as it serves as a diagnostic test with high sensitivity and good specificity [133]. Vitamin B12 also plays a crucial role in addressing specific challenges faced by patients with HTs, like the risk of anemia that may be heightened by coexisting autoimmune conditions like pernicious anemia or atrophic gastritis [134,135]. Research involving 130 patients diagnosed with autoimmune hypothyroidism revealed that 46% of them had vitamin B12 deficiency, which correlated with the presence of the disease [136]. Furthermore, patients with low vitamin B12 levels exhibited significantly higher levels of TPOAb, indicating a negative correlation between vitamin B12 and TPOAb antibodies [136]. Studies involving patients with AIT found that a considerable percentage had serum vitamin B12 levels below the reference range, although there was no significant correlation between B12 and TPOAb [137]. Moreover, a study comparing TgAb- or TPOAb-positive patients with healthy controls revealed a higher frequency of hemoglobin, iron, or vitamin B12 deficiency among patients [137], highlighting the need for further investigation into the risk of vitamin B12 deficiency in HTs patients. However, a recent systematic review and meta-analysis showed no difference between B12 levels in patients with HTs, hyperthyroidism, or subclinical hypothyroidism, but individuals with hypothyroidism had lower B12 levels compared to healthy adults [138]. This suggests that screening for anemia and B12 deficiency in patients with hypothyroidism is of importance. Further studies should focus on the association between vitamin B12 deficiency and AITD and other comorbid conditions.

## 5. Nutrition, Gut Microbiota, and Thyroid Function

The intestinal microbiota plays a crucial role in regulating metabolism and energy balance by extracting nutrients from food and producing metabolites that can influence host metabolic processes. Several studies have illustrated changes in microbiota composition following dietary modifications. The intestinal bacterial flora plays a crucial role in maintaining metabolic, nutritional, and even immunological homeostasis [139,140]. The hypothesis of a thyroid–gut axis is becoming increasingly concrete, suggesting that the gut microbiota, consisting of trillions of microorganisms inhabiting the gastrointestinal tract, may influence thyroid function through various mechanisms. In addition to influencing the absorption of minerals important for thyroid function (i.e., iodine, selenium, zinc, and iron), the microbiota is involved in the endogenous and exogenous metabolism of thyroid hormones. The presence of iodothyronine deiodinase (i.e., an enzyme that converts thyroxine into its active form, triiodothyronine, or into reverse T3 [rT3]) has been found in the intestinal wall, thus influencing total body T3 levels [2].

Exposure to endocrine disruptors (EDCs) can modify the composition of the gut microbiota, leading to changes in energy metabolism and weight gain. During the first 1000 days, breast milk represents one of the primary sources of EDCs exposure for newborns, as these compounds, metabolized and accumulated in the human body, have been detected in maternal milk, serum, hair, and placenta [3,141,142]. The entero-mammary circle is believed to facilitate the transfer of these compounds from the maternal bloodstream to infants via breast milk. Phthalates, including Di-2-ethylhexyl phthalate (DEHP), Di-isononyl phthalate (DINP), and Bisphenol A (BPA), are among the most common EDCs detected in breast milk [141,143,144,145]. Maternal exposure to EDCs can be influenced by various factors such as lifestyle, place of residence, occupation, and types of personal products used [141,142,146]. Studies have hypothesized that higher EDCs exposure can stimulate the synthesis of pathogenic microbial species [146,147], which may impact the development of thyroid disorders. A review by our group reported the role of the microbiota in metabolizing EDCs through a bidirectional interaction that leads to dysbiosis and alteration of pathways involved in the development of various metabolic diseases [141]. Dysbiosis and the consequent negative influence on the immune system and intestinal permeability can promote the development of inflammatory and autoimmune diseases, including thyroid diseases.

Diet plays a multifaceted role in the connection between the epigenome, thyroid function, and gut microbiota, exerting both direct and indirect effects on each of these systems [148,149]. Epigenetic modifications can influence the expression of genes involved in thyroid hormone synthesis, metabolism, and signaling pathways, thereby shaping thyroid function and hormone responsiveness [149,150]. The gut microbiota is known to produce metabolites that can influence epigenetic modifications in host cells, thereby exerting indirect effects on thyroid function. For example, short-chain fatty acids produced by gut bacteria have been shown to modulate histone acetylation and DNA methylation, which may impact thyroid hormone metabolism and signaling [151]. The methyl-donor nutrients folate and vitamin B12, which are rich in dark leafy greens, beans, peas, lentils, lemons, bananas, melons, and cereals, e.g., are involved in DNA methylation and chromatin remodeling, a key epigenetic mechanism [152]. It could be hypothesized that potentially consuming a diet rich in methyl-donor nutrients may promote optimal thyroid health. Similarly, dietary polyphenols found in fruits, vegetables, and tea have been shown to modulate histone modifications and can be used as a preventive recommendation for thyroid disorders [153]. Conversely, thyroid hormones can also influence the composition and function of the gut microbiota. Thyroid hormone receptors are expressed in the gut epithelium and immune cells, suggesting a direct role for thyroid hormones in gut physiology and immune function [154].

Supplementation with probiotics has exhibited positive impacts on thyroid hormone levels and thyroid function, as this approach may restore intestinal balance and promote the growth of beneficial microorganisms [155]. However, recent studies indicate the need for further research in the pediatric field regarding the interaction between microbiota and AITD [155]. Educating patients about the associations between gut and thyroid health can promote strategies to improve gut microbiota, especially in pregnant women and infants. Moreover, there is a strong need for further experimental studies to determine the potential benefits of nutritional interventions in HTs.

## 6. Conclusions

The importance of an adequate iodine intake with the diet is confirmed at all ages, while selenium supplementation must be carried out carefully after having corrected previous iodine insufficiency, as selenium has relationships with the development of autoimmune thyroiditis. Iron, vitamin D, zinc, and other micronutrients also play a role in thyroid function, but the studies currently available are insufficient for definite conclusions and recommendations. The main key points of current studies suggest that:-iodine is essential for thyroid hormone synthesis and beneficial for the prevention of thyroid disorders;-selenium may reduce thyroid autoantibodies and improve thyroid function, even though careful monitoring is required;-hypothyroid individuals should be assessed for anemia, iron, and vitamin B12 deficiency;-vitamin D supplementation might be beneficial for patients with Hashimoto thyroiditis;-the evaluation of zinc levels is recommended for subjects with thyroid enlargement; and-copper and magnesium should be studied for their potential protective role against thyroid cancer, and vitamin A for autoimmune thyroid disease.

Nutrition and gut microbiota can influence thyroid function by modulating immune responses, producing microbial metabolites, affecting nutrient absorption and epigenetic changes regulating thyroid hormone synthesis and metabolism, leading to the development of thyroid disease or protecting against it. Balanced nutrition is an essential source for maintaining healthy, optimal thyroid function and health. The diet–gut–thyroid axis should be considered for the possibility of modulating gut microbiota and thyroid health.

## Figures and Tables

**Figure 1 nutrients-16-02496-f001:**
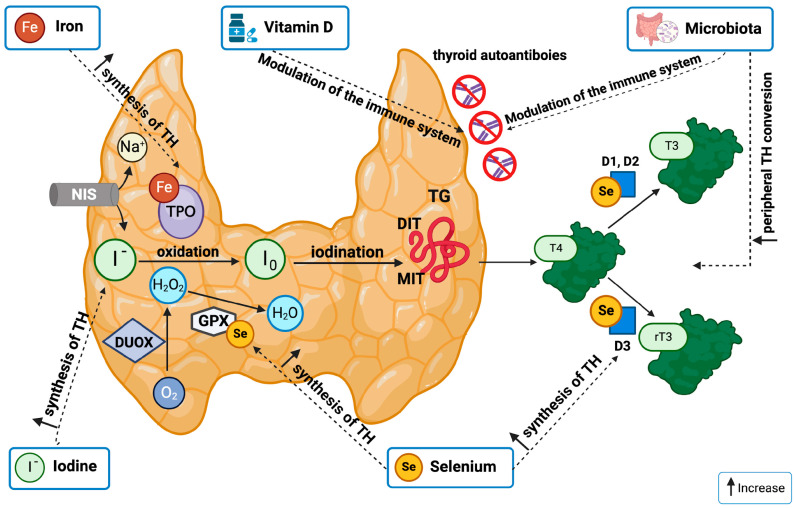
The overall effects of known nutrients and micronutrients on thyroid function. NIS: Na^+^ −Iodid Symporter (iodide uptake); DUOX: dual oxidase (H_2_O_2_ generation); TPO: thyroperoxidase (hemoprotein); TG: thyroglobulin (synthesis and storage protein); GPX: glutathione peroxidase (antioxidant defense); thyroid hormones (TH); monoiodotyrosine (MIT) and diiodotyrosine (DIT); D1, 2, 3: deiodinase (TH in−/activation). Created with BioRender.com, accessed on 19 April 2024.

**Table 1 nutrients-16-02496-t001:** Iodine intake requirements according to age according to the regulatory authorities.

Age	Adequate Intake (ug/Day) ^REF^ * for EFSA	Recommended Iodine Intake (ug/Day) ^REF^ ** for WHO
0–6 months old	-	90
7–12 months old	70	90
1–6 years old	90	90
7–10 years old	90	120
11–14 years old	120	120–150
15–17 years old	130	150
≥18 years old	150	150
During pregnancy	200	250
During lactation	200	250

REF, reference; * European Food Safety Authority Panel, EFSA J, 2014 [8]. ** World Health Organization and Food and Agriculture Organization of the United Nations. Vitamin and Mineral Requirements in Human Nutrition: Report of a Joint FAO/WHO Expert Consultation^.^ 2nd ed. World Health Organization; Geneva, Switzerland: 2004. [(accessed on 11 February 2024)] [9].

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
