# Peer review of "The Role of Nutrition on Thyroid Function"

_nutrients, 2024, doi:10.3390/nu16152496_

Round 1

Reviewer 1 Report

Comments and Suggestions for Authors

This narrative review explores the influence of nutrition and micronutrients on thyroid development and function, highlighting the diet-gut-thyroid axis. This review emphasizes the impact of micronutrients, such as iodine, selenium, iron, zinc, copper, magnesium, vitamin A, and vitamin B12, on thyroid health and discusses the role of gut microbiota in thyroid function.

Strengths

The manuscript provides an extensive review of the current knowledge on the relationship between nutrition and thyroid function, encompassing a wide range of micronutrients and their effects.

The review is well organized, with distinct sections covering different micronutrients and their specific roles in thyroid health.

Given the increasing interest in the impact of diet on endocrine health, this review addresses this significant and timely topic.

Weaknesses

This review summarizes existing knowledge without providing new insights or hypotheses that could advance the field.

The manuscript includes studies of varying quality, some of which may not meet rigorous scientific standards, potentially affecting the overall conclusions.

Some sections are more detailed and thorough than others, leading to an uneven depth of analysis across different micronutrients.

The methodology for the literature search is clearly described, adhering to systematic review principles. However, the criteria for selecting studies could be more stringent to ensure the higher quality and relevance of the included studies.

This review covers a broad range of micronutrients and their effects on thyroid function. 

Identify areas where evidence is lacking or conflicting, and suggest directions for future research.

Include more quantitative data or meta-analyses, where it is possible to provide a clearer picture of the magnitude of effects.

Ensure that each micronutrient has similar depth and clarity.

Propose new hypotheses or research questions based on the reviewed evidence.

Provide more detailed recommendations for clinicians and public health practitioners based on these findings.

Address minor grammatical errors and improve punctuation for better readability.

Include a more thorough assessment of the methodological quality of the included studies.

Incorporate quantitative data or meta-analyses where possible.

Ensure that all micronutrients are discussed with similar depth and clarity.

Propose new hypotheses or research questions based on the reviewed evidence.

Provide detailed recommendations for clinicians and public health practitioners.

Comments on the Quality of English Language

Minor flaws

Author Response

Comment: This narrative review explores the influence of nutrition and micronutrients on thyroid development and function, highlighting the diet-gut-thyroid axis. This review emphasizes the impact of micronutrients, such as iodine, selenium, iron, zinc, copper, magnesium, vitamin A, and vitamin B12, on thyroid health and discusses the role of gut microbiota in thyroid function.

Strengths
The manuscript provides an extensive review of the current knowledge on the relationship between nutrition and thyroid function, encompassing a wide range of micronutrients and their effects.
The review is well organized, with distinct sections covering different micronutrients and their specific roles in thyroid health.
Given the increasing interest in the impact of diet on endocrine health, this review addresses this significant and timely topic.

Weaknesses
This review summarizes existing knowledge without providing new insights or hypotheses that could advance the field.
The manuscript includes studies of varying quality, some of which may not meet rigorous scientific standards, potentially affecting the overall conclusions.
Some sections are more detailed and thorough than others, leading to an uneven depth of analysis across different micronutrients.
The methodology for the literature search is clearly described, adhering to systematic review principles. However, the criteria for selecting studies could be more stringent to ensure the higher quality and relevance of the included studies.
This review covers a broad range of micronutrients and their effects on thyroid function. 
Identify areas where evidence is lacking or conflicting, and suggest directions for future research.
Include more quantitative data or meta-analyses, where it is possible to provide a clearer picture of the magnitude of effects.
Ensure that each micronutrient has similar depth and clarity.
Propose new hypotheses or research questions based on the reviewed evidence.
Provide more detailed recommendations for clinicians and public health practitioners based on these findings.
Address minor grammatical errors and improve punctuation for better readability.
Include a more thorough assessment of the methodological quality of the included studies.
Incorporate quantitative data or meta-analyses where possible.
Ensure that all micronutrients are discussed with similar depth and clarity.
Propose new hypotheses or research questions based on the reviewed evidence.
Provide detailed recommendations for clinicians and public health practitioners.

Response: We appreciate the reviewer’s feedback and insightful suggestions to improve our review. We have updated the paper with recent systematic reviews and meta-analyses. To balance the manuscript we have added headlines to separate main sections with respect to additional information. As the reviewer recommended for each section we added evidences and suggested directions for further research, as well as recommendations for health practitioners.  Thank you for taking the time to give us your valuable feedback.

Reviewer 2 Report

Comments and Suggestions for Authors

The authors present a comprehensive review of literature going back to 1986 describing the effects of micronutrients (and the gut microbiota) on thyroid function.  I appreciate that they took pains to point out when different articles on the same topic came to different conclusions and I think their conclusions are scientifically sound.  A few suggestions:

1. Line 136:  needs to be reworded as it suggest that iodized salt is not used widely in the US which I know is not true

2. Line 409-410: they need to make clear that the evidence that selenium helps with hair loss associated with hypothyroidism was based as far as I could tell on a single case report

3.  I would say that zinc supplementation MIGHT (not CAN) be useful in hypothyroid patients.

4. Lines 449-459:  this paragraph is poorly written and the key message needs to be made clearer

5. Lines 599-621:  It is not clear why this study (reference 139) is discussed in such depth or included at all, since it does not have a clear relevance to thyroid metabolism. I would suggest deleting.

7. Conclusion:  I agree with what the authors have said here but given the large amount of material they covered, I would expand this section to include a listing of 6-8 key points for which they found the strongest evidence in the literature

Author Response

The authors present a comprehensive review of literature going back to 1986 describing the effects of micronutrients (and the gut microbiota) on thyroid function.  I appreciate that they took pains to point out when different articles on the same topic came to different conclusions and I think their conclusions are scientifically sound.  A few suggestions

Comment 1: Line 136:  needs to be reworded as it suggest that iodized salt is not used widely in the US which I know is not true

Response 1: Thank you for pointing this important aspect out. We agree with the comment and have deleted this part from the text to avoid misunderstanding.

Comment 2: Line 409-410: they need to make clear that the evidence that selenium helps with hair loss associated with hypothyroidism was based as far as I could tell on a single case report

Response 2: In our version of the document lines 409-410 are related to the zinc paragraph. Thank you for suggesting clarifying details about the role of zinc in hair loss.

Comment 3: I would say that zinc supplementation MIGHT (not CAN) be useful in hypothyroid patients.

Response 3: We agree with the reviewer’s insightful comment about zinc supplementation, and changed the line for better clarification.

Comment 4: Lines 449-459:  this paragraph is poorly written and the key message needs to be made clearer

Response 4: We appreciate your suggestion. We rewrote and added more information to this section.

Comment 5: Lines 599-621:  It is not clear why this study (reference 139) is discussed in such depth or included at all, since it does not have a clear relevance to thyroid metabolism. I would suggest deleting.

Response 5: We agree with the reviewer’s comment about deleting this part from the text.

Comment 6: Conclusion:  I agree with what the authors have said here but given the large amount of material they covered, I would expand this section to include a listing of 6-8 key points for which they found the strongest evidence in the literature

Response 6: Thank you for your suggestion of key points in the conclusion part that can improve it.